Urban waste disposal explains the distribution of Black Vultures (Coragyps atratus) in an Amazonian metropolis: management implications for birdstrikes and urban planning

de Araujo Giase M. 1
Peres Carlos A. c.peres@uea.ac.uk 2 3
Baccaro Fabricio B. 4
Guerta Rafael S. 1 5
1 Wildlife Management Center, Eduardo Gomes International Airport of Manaus , Manaus , Amazonas , Brazil
2 Centre for Ecology, Evolution and Conservation, School of Environmental Sciences, University of East Anglia , Norwich , Norfolk , United Kingdom
3 Departamento de Sistemática e Ecologia, Centro de Ciências Exatas e da Natureza, Universidade Federal da Paraíba , João Pessoa , Para , Brazil
4 Departamento de Biologia, Universidade do Amazonas , Manaus , Amazonas , Brazil
5 Departamento de Zoologia, Universidade Federal do Pará , Belém , Pará , Brazil
Martinelli Luiz
Electronic publication date: 2018 Sep 14
Publication date: 2018
Volume: 6
Electronic Location ID: e5491
Received 2018 Mar 13; Accepted 2018 Jul 30
Copyright: ©2018 Araujo et al.
Copyright year: 2018
Copyright holder: Araujo et al.
License: This is an open access article distributed under the terms of the Creative Commons Attribution License, which permits unrestricted use, distribution, reproduction and adaptation in any medium and for any purpose provided that it is properly attributed. For attribution, the original author(s), title, publication source (PeerJ) and either DOI or URL of the article must be cited.
License URL: https://creativecommons.org/licenses/by/4.0/

Keywords: Aviation safety, Human-wildlife conflicts, Urban ecology, Cathartidae, Coragyps atratus, Amazonian cities

Funding: Programa Fauna Nos Aeroportos Brasileiros Centro de Apoio ao Desenvolvimento Tecnológico (CDT)/Universidade de Brasília (UnB) Empresa Brasileira de Infraestrutura Aeroportuária (INFRAERO) Coordenação de Aperfeiçoamento Pessoal de Nível Superior (CAPES) Global Innovation Initiative grant GII-111 Conselho Nacional de Desenvolvimento Científico e Tecnológico (CNPq) #309600/2017-0 This study was funded by ‘Programa Fauna Nos Aeroportos Brasileiros’. During this study Rafael Guerta and Giase Araujo received a fellowship from the partnership between the Centro de Apoio ao Desenvolvimento Tecnológico (CDT)/Universidade de Brasília (UnB) and the Empresa Brasileira de Infraestrutura Aeroportuária (INFRAERO). Rafael Guerta was funded by a PhD studentship from the Coordenação de Aperfeiçoamento Pessoal de Nível Superior (CAPES). Carlos Peres was funded by a Global Innovation Initiative grant (GII-111). Fabricio Baccaro is continuously supported by Conselho Nacional de Desenvolvimento Científico e Tecnológico (CNPq) grant #309600/2017-0. The funders had no role in study design, data collection and analysis, decision to publish, or preparation of the manuscript.

==============================
Collision rates between aircraft and birds have been rising worldwide. The increases in both air traffic and population sizes of large-bodied birds in cities lacking urban planning result in human-wildlife conflicts, economic loss and even lethal casualties. Black Vultures (Coragyps atratus) represent the most hazardous bird to Brazilian civil and military aviation on the basis of their flight behavior, body mass and consequently physical damage to aircraft following collisions. This study investigated how storage apparatus and type of organic residue discarded in public street markets modulate the spatial distribution and abundance of urban Black Vultures in the largest city in the Amazon (Manaus, Brazil). We estimated Black Vulture abundance in relation to the type of solid human waste (animal or plant), the type of waste storage containers and market sizes in terms of the number of vendor stalls at 20 public markets. We also visually quantified the abundance of Black Vultures in urban markets in relation to air traffic. Our results suggest that urban solid waste storage procedures currently used (or the lack thereof) are related to the occurrence and abundance of Black Vultures. Moreover, storage type and the proportion of animal protein (red meat and fish) within rubbish bins directly affects foraging aggregations in vultures. We recommend that policymakers should invest more efforts in building larger and more resistant closable waste containers to avoid organic solid waste exposure. We also identified five outdoor markets as urgent priorities to improve waste disposal. Finally, our waste management guidelines would not only reduce aviation collision risks but also benefit human health and well-being in most cities.

Introduction

Collision rates between aircraft and birds, widely referred to as birdstrikes, have been escalating worldwide (Allan, 2000; Dolbeer, 2011; Dolbeer et al., 2016). Some of the known reasons for such increases include both the burgeoning aviation industry leading to much higher air traffic (Mendonça, 2009) and increases in bird population sizes in cities lacking urban planning (Bastos, 2000; Dolbeer & Eschenfelder, 2003; Dolbeer & Seubert, 2009). In some cases, birdstrikes can result in severe damage to aircrafts, less frequently leading to deaths of both crew and passengers (Richardson & West, 2000; Allan, 2002; Thorpe, 2003; Thorpe, 2005; Thorpe, 2016). As a result, bird population monitoring and wildlife control measures have increasingly become a glaring concern for aviation agencies (Federative Republic of Brazil, 2012; ANAC, 2014; AAA, 2016).

Although birds are the main taxonomic group involved in aircraft accidents (Cleary, Dolbeer & Wright, 2005), the degree to which any specific bird species can be considered a hazard to aviation depends on the body size, habitat and behavior of individual bird populations (Dolbeer, Wright & Cleary, 2000; DeVault et al., 2011). In Brazil, most birdstrike collisions result from large-bodied urban raptors (e.g., vultures, hawks and falcons) belonging to the families Cathartidae, Accipitridae, and Falconidae (CENIPA, 2016). Further, over the last three decades, vultures (Cathartidae, Lafresnaye, 1839) were ranked as the most hazardous birds to Brazilian urban aviation on the basis of on their flight behavior, large size and consequently physical damage to aircrafts following collisions (Serrano et al., 2005; Novaes & Alvarez, 2010; Novaes & Alvarez, 2014; CENIPA, 2016). For instance, a single collision involving two Turkey Vultures (Cathartes aura) in 2012 near the Manaus International Airport (MAO) resulted in engine failure and financial losses of ∼US$ 750,000 (CENIPA, 2017).

The spatial distribution of birds in urban environments is related to several factors including the presence of food resources and shelters (roosting sites) as, for example, in Black Vultures (Coragyps atratus) (Novaes & Cintra, 2013). It is therefore possible to reduce birdstrike probability by simply modifying and monitoring the environment surrounding airports and rendering them less attractive to birds as foraging habitats (Blackwell et al., 2009; Martin et al., 2011). As the majority of birdstrikes occur both within and very near airport landing and take-off areas (Dolbeer, 2006; Dolbeer et al., 2016), it is crucial to identify (i) what urban features are attractive to birds, and (ii) which species are most often directly related to collisions, both of which are crucial to urban planners to improve aviation safety.

The Black Vulture is an ubiquitous and often abundant species in Brazilian cities (Sick, 1997). They are easily identified by the absence of head and neck feathers, which may assist with hygiene after eating, since their diet is frequently comprised of decomposing animal carcasses and other organic human waste (DeVault et al., 2016). Black Vultures are highly successful opportunist scavengers and exhibit gregarious behavior, actively using manmade structures to both forage and roost (Novaes & Cintra, 2013). They also frequently take advantage of wind thermals to gain flight elevation and save energetic expenditure while soaring (Freire et al., 2015).

In this study, we examine how the storage apparatus and type of organic residue discarded in public street markets influence the distribution and abundance of urban Black Vultures, the main bird species involved in birdstrike risk at large Amazonian cities. During an observational study, we estimated Black Vulture abundance in relation to the type of solid human waste (animal or plant), the type of waste storage containers and market sizes in terms of the number of vendor stalls at 20 public outdoor markets in Manaus. We also visual examined the abundance of Black Vultures in urban markets in relation to the main hubs of air traffic. We further recommend simple management and storage measures to control the local abundance of this species. This low-cost set of suggested actions can be rapidly implemented to reduce the probability of avian collisions with aircraft, and they are also expected to work in other cities facing similar problems.

Materials & Methods

Study area

Fieldwork was conducted in both urban and suburban areas of Manaus (03°80′S, 60°01′W), in Central Brazil Amazonia. Manaus is the largest and most populated Amazonian city, spans an area of 377.4 km2 and is surrounded by rainforest to the north and east, and the Negro and Amazon rivers to the south and west. Mean annual temperature in Manaus is 30 °C and the climate is classified as tropical humid (Loureiro, Carlos & Lamberts, 2001). Mean annual precipitation is 2,195 mm with a rainy season between December and May, and a dry season between June and November (Loureiro, Carlos & Lamberts, 2001). The city has grown rapidly since it became a duty-free port and hub of the Brazilian manufacture industry, doubling its population size in recent decades. In 1991, Manaus had approximately 1 million inhabitants, whereas the latest census (2017) recorded a population size of 2.13 million (IBGE, 2017), representing a 113% increase over three decades.

Bird counts

Manaus has 35 open-air markets registered by the urban administration as regular legal markets (SUBSEMPAB, 2018). These public markets are distributed throughout the entire city and are used to commercialize several types of local products including red meat, fish, fruits and vegetables. We monitored 20 open-air public markets distributed across the main city neighborhood sectors (or bairros). Our aim was to estimate black vulture population size over the entire city area. Our sampling design covered ∼60% of all open-air markets in Manaus. This wide variety of markets allowed us to examine different forms of organic waste storage. Monitoring occurred between 18 November 2014 and 20 February 2015. Sampling consisted of five visits to each street market from 09:00 h to 16:00 h, the time interval corresponding to the peak activity period for Black Vultures (e.g., Avery et al., 2011; Novaes & Cintra, 2015). Each visit lasted between 10 and 15 min and was conducted by one observer (G.M.A.) in two steps. First, the observer inspected the entire area of the markets in order to locate and count all individual vultures present in a vehicle driven by a motorist at a constant slow velocity of <20 km/h. The markets were located along linear streets and in most markets the garbage disposal was concentrated in a single main location per market, thereby facilitating vulture counts. After carefully inspecting the entire market area, the observer left the vehicle to count from a reasonably short distance (beyond escape responses) all individuals that were on the ground. The detection radius was ∼30 m. Subsequently, all vultures soaring overhead and above the market were also counted. To avoid double counting, only one observer recorded local vulture abundance. At each visit, the abundance and location of all Black Vultures were recorded, as well as their main behavioral activity pattern (i.e., resting, flying, foraging, and social interactions).

Waste storage

A classification of waste type (red meat, fish, fruits and vegetables) was assigned to any exposed organic waste observed during each bird census period using a visual estimation method. A standardized estimate of the proportion of each waste type was conducted by the same observer (G.M.A.) at all street markets. Waste storage was categorized as “without container”, whenever any given market lacked any type of rubbish bin facilities or specific storage containers, thereby resulting in waste being discarded haphazardly at open-air sites. Markets containing rubbish bin facilities were classed into two types: “small house” and “container” markets (Fig. 1).

Figure 1 Types of rubbish bins normally present in street markets in Manaus, Brazil.

(A) “Small house” rubbish bin without doors at Armando Mendes street market. (B) Closed “small house” rubbish at Bairro da Paz street market. (C) “Container” rubbish bin at Ceasa street market. (D) “Container” rubbish bin at Compensa street market. (E) “Container” rubbish bin at Coroado market. (F) “Container” rubbish bin at Feira do Produtor street market. Photo credit: Giase M. de Araújo.

“Small house” rubbish bins were of a rectangular shape, made of brick masonry with a door entrance (but often lacking a door), and were used to protect any solid waste from wind and rain. Those rubbish bins were made by local vendors from each street market (Figs. 1A and 1B). “Container” rubbish bins, on the other hand, consisted of metal box frames covered by a heavy protective lid. The containers usually provided effective waste protection, but in all cases the metal lid was removed or remained open at all times in order to facilitate waste disposal (Figs. 1C–1F). Although these two types of rubbish containers were the most common in street markets in Manaus and were cleaned on a daily basis by rubbish collectors hired by the public waste management service, they lacked any regular repair or maintenance service.

Historical birdstrike records and sighting reports

All collisions events with vertebrate species (bird, bats, terrestrial reptiles and mammals larger than 1 kg) found or observed within an airport operation area are reported to the federal wildlife strike management agency (Aeronautical Accidents Investigation and Prevention Center; hereafter, CENIPA) together with details describing the event and the species, according to law 12.725/2012 (CENIPA, 2017). Furthermore, a resolution (No. 466/2015) from the National Environmental Council issued the first risk assessment methodology applied to wildlife within the aviation sector context. This resolution defines and categorizes each incident according to the following details: (i) crewmembers or ground airport operators observes a collision event; (ii) maintenance personnel identifies evidence or damage caused by the collision with the aircraft; and (iii) animal remains are found within 50 m of runway or taxiway boundaries, or within 300 m from a runway threshold, unless qualified technical personnel identifies the cause of death of the animal as unrelated to a collision event. The resolution also defines air strikes as event occurring generally outside aerodrome boundaries, in one of the following flight phases: take-off, climb, cruise, low-level navigation, descent, approach (landing) and during in-transit inspections. Additional details can be found at the Brazilian Annual Wildlife Strike Summaries available at the CENIPA website (http://www2.fab.mil.br/cenipa/index.php/estatisticas/risco-da-fauna). We extracted information on birdstrikes and number of aircraft movements for the two main airports of Manaus on the basis of systematic records spanning three consecutive years (CENIPA annual reports: 2013–2015). To provide a comparative perspective, all data and collision analyses are conventionally reported for each set of 10,000 aircraft movements (flights). We used this information together with airport and street markets spatial data to understand spatio-temporal patterns of birdstrikes in Manaus.

Data analysis

Since our main response variable consists of discrete count data based on five visits to each market, we used generalized linear mixed models (GLMM) with a Poisson distribution of residuals. The number of vultures observed during each market survey was our dependent variable and market size (estimated by the number of red meat/fish boxes in the market), proportion of solid animal waste (red meat and fish/total waste), and presence/absence of rubbish bins were the predictor variables in the model. We build separate models to explain vulture abundance on the ground or flying, as well as total abundance. We counted only vultures flying at relatively low heights in the immediate proximity to markets, so that individuals soaring in thermals at high elevations were excluded. To account for our spatially nested design, we included market as a random variable in the models. To validate the model fit, we compared the Akaike’s Information Criterion (AIC) of the GLMM with the respective AIC of the null model (intercept only + random variable). The full GLMM model was selected, when the AIC difference between the full and null model was greater than 2 (Burnham & Anderson, 2002). We report the marginal and the conditional R2 for the GLMM model. Marginal R2 provides the variation explained by the fixed variables only, while the conditional R2 gives the variation explained by the fixed and random effects in the model (Nakagawa & Schielzeth, 2013). We also performed residual analysis to verify the adequacy of model predictions. We used partial graphs to present the GLMM results in simple 2D plots. The graph shows expected values of the dependent variable and expected values of the target independent variable if all other independent variables in the analysis were maintained at their median values (Breheny & Burchett, 2017). All analyses were carried out in R 3.2.3 (R Development Core Team, 2018).

To define the order of priority of street markets related to any possible Black Vulture population control and waste storage management measures, with a view of reducing the probability of birdstrikes, we performed two procedures. First, we superimposed street market locations and the numeric abundance of vultures found at each market with the mainstream aircraft routes within the entire airspace of the city of Manaus. The vulture abundance was scaled to the percentage of animal matter in waste residues at different markets. Manaus has three aerodromes, two of which serving high daily traffic of both passengers and cargo: Eduardo Gomes International Airport of Manaus (ICAO: SBEG), Ponta Pelada Air Force Base (SBMN), and Flores Airport (SWFN). As a private owner operates the much smaller Flores Airport, data on birdstrike and aircraft movements are unavailable. However, the SWFN airport is now used only by small single-engine airplanes and hosts fewer flights on a daily basis compared to the other two aerodromes. Therefore, our analysis was primarily restricted to the SBEG and SBMN aerodromes, which collectively hosted 158,223 flights over the 3-year study period. Second, we created a composite index (DI) of distance to the nearest airports by dividing the airport size (mean number of aircraft flights per year) by the square-root of the linear (vulture flight) distance from each market to that airport. We then merged the information about airport distance with the abundance of Black Vultures of individual surveys/visits, to create a rank of priority for Black Vulture population control.

Results

Average Black Vulture abundance varied considerably across the 20 urban markets surveyed during the sampling period (Table 1). The absence of vultures was detected in half of the study markets, whereas at another 10 street markets the average abundance varied from 28 at Feira do Walter Rayol to 810 individuals at Feira da Panair (sd = 48.40). Two vulture concentration zones could be clearly detected in Manaus, and these are located in the extreme north portion of urban area (coinciding with three outdoor street markets) and the main fishing port of the city located near the downtown commercial area, where fish discards are common (Fig. 2).

Table 1 Name, size and geographic coordinates of street markets monitored in Manaus, Brazil.

Name, size (in terms of the number of vendor stalls) and geographic coordinates of street markets monitored in Manaus, Brazil, and the type of rubbish bin (whenever available) and the corresponding Black Vulture abundance.

Street market	Geographic coordinates	Market size	Rubbish bin	Behavior	Total abundance	
				Flying	Grounded		
Feira do Bairro da Paz	3°3′24.72″S; 60°1′56.63″W	5	Small House	0	0	0	
Feira João Sena	3°4′37.33″S; 60°2′18.35″W	21	Absent	0	0	0	
Feira Dorval Porto	3°6′47.46″S; 60°1′24.77″W	8	Absent	0	0	0	
Mercado Senador Cunha Melo	3°7′20.99″S; 60°1′35.40″W	20	Absent	0	0	0	
Feira da Manaus Moderna	3°8′29.05″S; 60°1′19.91″W	137	Absent	78	82	160	
Feira da Panair	3°8′45.43″S; 60°0′37.88″W	137	Container	445	365	810	
Mercado do Walter Rayol	3°8′4.86″S; 60°0′32.34″W	38	Container	23	5	28	
Mercado Maximinio Correa	3°7′13.79″S; 60°0′54.86″W	3	Absent	0	0	0	
Feira do Porto da Ceasa	3°8′3.10″S; 59°56′26.07″W	21	Container	0	186	186	
Feira do Japiim	3°6′44.19″S; 59°58′52.70″W	9	Absent	0	0	0	
Feira Santo Antônio	3°7′11.92″S; 60°2′39.17″W	19	Container	0	31	31	
Feira da Compensa	3°6′27.24″S; 60°3′18.58″W	40	Container	35	192	227	
Feira da Raiz	3°7′27.72″S; 59°59′42.40″W	5	Container	0	0	0	
Feira do Coroado	3°5′2.33″S; 59°58′48.71″W	7	Container	0	116	116	
Feira da Armando Mendes	3°5′24.78″S; 59°56′35.55″W	20	Small House	0	0	0	
Feira do Mutirão	3°2′44.91″S; 59°56′36.58″W	46	Absent	53	250	303	
Feira do Produtor	3°2′11.32″S; 59°56′26.66″W	14	Container	0	0	0	
Feira da Cidade Nova	3°1′53.62″S; 59°59′5.64″W	10	Container	44	106	150	
Feira da Nova Cidade	3°0′17.29″S; 59°58′32.69″W	3	Container	0	1	1	
Feira do Mundo Novo	3°2′23.31″S; 59°59′53.93″W	10	Absent	0	0	0	

Figure 2 Map of the Manaus metropolitan area and the street markets surveyed in this study.

Red circles and blue squares represent street markets where vultures were present or absent, respectively. Red circle sizes are proportional to the local abundance of vultures recorded at each street market. Bold straight lines indicate the runway location of the two main airports in Manaus: Eduardo Gomes International Airport of Manaus (SBEG) and Ponta Pelada Air Force Base (SBMN). White arrows show the main aircraft paths for either take-off or approach according to aeronautical charts. Image credit: Google, TerraMetrics, 2018.

The full GLMM models provided a better fit compared to the respective null model (total abundance ΔAIC = 14.25, abundance on the ground ΔAIC = 14.55 and abundance of flying vultures ΔAIC = 2.07). The models that included vultures that were either perched or on the ground provided a much better fit, compared with the model based solely on flying Black Vultures (Table 2). The fixed variables explained ∼63% of the variation in total vulture abundance and those on the ground, but a reasonable part of the variation in Black Vulture abundance were also associated with idiosyncrasies of each market (Conditional R2 ≈ 0.88 in both models). The presence or absence of adequate rubbish bins and the proportion of animal protein (i.e., red meat and/or fish) available in the solid waste discarded within each street market were strongly positively related with the abundance of Black Vultures in both models (Table 2). Market size in terms of the number of vendor stalls, which may be used as a proxy for amount of waste discarded was, however, unrelated with the abundance of Black Vultures (Table 2). The abundance of flying Black Vultures was more related to the size of the market, than to the presence/absence of rubbish bins and the proportion of animal waste. However, the model explained only ∼16% of the abundance of vultures in flight (Table 2). This indicates that the number of Black Vultures scales similarly to food availability in markets with and without adequate rubbish bin facilities, where waste is discarded haphazardly, but that disposing of organic rubbish into bins decreased vulture abundance by ∼25% (Figs. 3A and 3B). The effect of rubbish bin use (either a container or a small house) was apparently equivalent even when container bins lacked lids or other means of effective cover.

Figure 3 Partial residual plots of the effects of the proportion of animal protein contained in discarded organic solid waste for each street market on number of vultures recorded at each survey.

Partial residual plots of the effects of the proportion of animal protein (red meat or fish) contained in discarded organic solid waste for each street market, where rubbish containers were either present (green circles, and green 95% CI region) or absent (red circles, and red 95% CI region) on total number of vultures (A) and number of vulture at the ground (B) recorded at each survey. The effects of market size (non-significant) and market identity (random variables) were both held constant.

Table 2 Summary of the generalized-linear mixed models results of Black Vultures abundance related to presence or absence of rubbish bins, proportion of animal protein and market size.

Marginal R2 values account for the variance explained by fixed variable only, while conditional R2 values represent the variance explained by both, fixed and the random factor. Market were declared as random factor in all models.

Dependent variable	R2 marginal	R2 conditional	Fixed factor	b	P-value	
Total vulture abundance	0.628	0.879	Presence or absence of rubbish bins	2.162	0.006	
			Proportion of animal protein	2.296	0.005	
			Market size	1.152	0.116	
Vulture abundance on the ground	0.644	0.877	Presence or absence of rubbish bins	1.986	0.018	
			Proportion of animal protein	2.890	0.001	
			Market size	0.781	0.290	
Abundance of vultures in flight	0.162	0.162	Presence or absence of rubbish bins	1.103	0.391	
			Proportion of animal protein	1.460	0.319	
			Market size	2.727	0.051	

The vast majority of collisions between birds and aircraft in Manaus occurred during take-off, approach and landing flight maneuvers (amounting to ∼73% of all 101 strikes recorded during the three years of CENIPA monitoring). Comparing the two Manaus airports, the largest airport (Eduardo Gomes International Airport) exhibited the highest number of reported animal collisions, although the rate of collisions (per aircraft flight) was higher at the Ponta Pelada Air Force Base (Table 3). This result can be explained by the spatial location of some street markets within the city. Only the fairly small street market of Feira da Cidade Nova is situated very near the main flight paths for either approach or take-off aircraft routes from the Eduardo Gomes International Airport and there were no records of Black Vultures at this market during the monitoring period (Fig. 2). On the other hand, four street markets containing large numbers of vultures (Feira da Panair, Feira da Manaus Moderna, Mercado Walter Rayol, and Feira da Ceasa) are situated very close to the approach and take-off paths of military aircraft using the Ponta Pelada Air Force Base (Figs. 2 and 4). In total, we recorded 1184 vulture sightings during our surveys at those four street markets combined, which represented 58.9% of all vultures sighted across all 20 street markets surveyed.

Table 3 Number of collisions, aircraft movements and collision index in different years for the two main airports in the city of Manaus, Brazil.

Number of collisions, aircraft movements and collision index (number of collisions per 10,000 flights) in different years for the two main airports in the city of Manaus. Flight movement data from the Flores Airport was unavailable.

Airport	Year	Number of collisions	Flights	Collision index	
Eduardo Gomes (SBEG)	2013	42	51,234	8.2	
	2014	26	53,752	4.84	
	2015	17	37,782	4.5	
Ponta Pelada (SBMN)	2013	3	5,702	5.26	
	2014	4	5,342	7.49	
	2015	9	4,411	20.4	

Figure 4 Mean number of Black Vultures in each urban market surveyed.

Mean number of Black Vultures represented by horizontal bars in each urban market surveyed. Horizontal lines represents the standard deviation of Black Vultures abundance of each market. The color gradient represents the composite index (DI) of distance to the nearest airport created by dividing the airport size (mean number of aircraft flights per year) by the square-root of the linear (vulture flight) distance from each market to that airport.

Discussion

Overall, it is evident that the waste storage system in urban street markets in Manaus and many other Amazonian cities is inadequate. For example, 40% of all monitored street markets lacked any waste storage facilities. Our results indicate that the procedures currently used (or the lack thereof) to store solid organic waste are clearly related to the occurrence and abundance of Black Vultures in the largest Amazonian city. Further, our results suggest that waste storage infrastructure and the proportion of animal protein waste discarded into rubbish bins are direct drivers of the main foraging area for urban vultures. Larger numbers of vultures were consistently found at sites lacking storage facilities. In particular, we note that just the presence of a rubbish bin, even if it lacks proper protective cover, was enough to prevent large vulture aggregations. Also, the number of vultures recorded within street markets was related to the proportion of fresh and/or decomposing animal waste available. In addition to affecting vulture occurrence, which induces higher probabilities of birdstrike collisions, poorly managed organic waste can lead to a series of detrimental environmental and epidemiological consequences, including accumulation of microbial pathogens, proliferation of disease vectors, and both soil and water contamination (Moore, Gould & Keary, 2003; Penrose et al., 2010).

The availability and accessibility to food resource are known as key determinant factors in shaping the habitat use and spatial distribution of cathartids (Jones, 2005; Novaes & Cintra, 2015). Also, the total amount of food resources available, which is a key attractant to vultures, is one of the main ecological factors that affect the fine-scale spatial distribution of vultures (Parmalee, 1954; DeVault et al., 2005; Novaes & Alvarez, 2010; Novaes & Cintra, 2013). Black Vultures use vision as the main sensory mechanism to find food resources (Houston, 1986), being able to detect food sources within a 90-km radius (Novaes & Cintra, 2015). Our results suggests that Black Vultures may locate the markets during low height flying based on market size, but further select and forages in markets with higher proportion of animal waste. Our results therefore corroborate previous evidence in showing a relationship between animal solid waste availability and local abundance of Black Vultures in large metropolitan areas where food waste is inadequately managed and discarded at open-air sites.

Organic waste made available in urban street markets is clearly a critical food source for Black Vultures. Consequently, the main population control strategy for this species is to eliminate or minimize the availability of organic waste produced every day by several human activities. Our results suggest that, given a lack of financial or human resources to deal with waste management properly, decision makers should prioritize actions targeting the treatment and management of waste of animal origin. Simple management protocols, such as disposing of any waste into small container bins, are probably sufficient to be highly effective. Furthermore, environmental education campaigns focused on market vendors should be implemented in order to increase their general awareness that waste accumulation can result in direct and indirect negative effects on both human health and aviation safety (Bastos, 2000; Araujo, Terán & Guerta, 2015). During our visits to the street markets, we noticed that even in markets with rubbish containers, local vendors still throw away organic solid waste on the street around containers. In fact, most street markets in Manaus exhibit extremely poor hygienic conditions in terms of organic waste scattered across neighboring streets. In addition, many rubbish containers present in some markets lacked proper cover. Therefore, solid waste stored within containers lacking a lid, remains available and can be easily found by vultures. Black Vulture relies primarily on vision to detect and locate food resources (Coleman & Fraser, 1989; Lisney et al., 2013). In street markets without a rubbish bin, any organic waste is more exposed and available to urban necrophages (including vultures, rats and feral dogs and cats) compared to markets using containers, thereby attracting more scavengers even if a smaller amount of organic waste is disposed. Likewise, even when waste was stored inappropriately into open containers lacking adequate cover was apparently enough to prevent large Black Vulture aggregations in small markets producing less organic waste.

Only 10% (2/20) of all street markets in our study area provided proper waste storage ‘small housing’ facilities that were inaccessible to urban scavengers. The lack of adequate infrastructure and efficient waste collecting services fuel a larger vulture population, and consequently inflate birdstrike probability risk in any large urban center in the tropics. Therefore, it is extremely important that policymakers invest more efforts in building larger and more resistant waste containers to avoid disease proliferation and organic pollution. In a financially restricted context, however, five street markets should be prioritized —Feira Cidade Nova, Feira da Panair, Feira da Manaus Moderna, Mercado Walter Rayol, and Feira do Ceasa—as they are located near take-off and landing routes of the two most important Manaus airports. Furthermore, intensive campaigns towards improving environmental awareness and changing local behavior and attitudes towards correctly storing market organic waste are complementary strategies. Finally, installation of adequately covered containers combined with more regular waste collecting services represent relatively inexpensive and easy alternatives to control Black Vulture hyper-abundance in large Amazonian cities like Manaus, as well as reducing the risk of birdstrikes.

Conclusion

The mismanagement of human waste derived from outdoor markets in the Amazon’s largest city induces the occurrence and local abundance of highly mobile urban scavengers, such as Black Vultures, thereby overinflating the risk of aviation accidents. Simple and low-cost measures to enhance waste disposal can reduce the size of local aggregations of avian scavengers, not to mention other urban ecosystem benefits in terms of urban planning, service provision and sanitation. Although urban waste disposal explains the spatial distribution of Black Vultures in large metropolitan areas such as Manaus, an efficiently adequate and frequent solid waste collecting service would not be enough to effectively control and reduce vulture populations in the median term. The amount of organic solid waste discarded, subsequently becoming readily available, provides ample trophic subsidies to the resident Black Vulture population in Amazonian towns and cities. An appropriate waste management protocol would only suppress urban scavenger populations if this is executed as part of a long-term policy taking into account multiple social agents linked to both waste production and management.

Supplemental Information

Supplemental Information 1 Black Vulture abundances, types of solid human waste and waste storage containers at 20 street markets in Manaus, Amazonas, Brazil

Date. start and end of surveys, weather conditions, names of street markets and their neighbourhoods, distance index, % fish and red meat, count data about presence/absence of Black Vultures and their behaviour, types of solid human waste and waste storage containers at 20 street markets in Manaus, Amazonas, Brazil.

Click here for additional data file.

We would to thank Weber G. Novaes for his crucial role in overall project design and development. We also thank Anderson Moreira da Silva and Diego Ribeiro Lima for their competent field assistance. Giase M. de Araujo thanks to Rosangela Melo de Araujo e Valdir Silva de Araujo for all the support. We are very grateful to all the wildlife management team from International Airport of Manaus Eduardo Gomes for their useful help during fieldwork and the administrative support. Finally, we thank the INFRAERO’s National Environmental Sector on their support, revision and constructive criticism. Opinions expressed here do not necessarily reflect current Civil Aviation National Agency (ANAC) of Brazil’s policy governing the control of wildlife near airports. In Memorium: we dedicate this paper to the vivid memory of our beloved friend, avian ecologist and coauthor, Rafael Guerta, who tragically passed away recently—a heartbreaking loss to us all. He leaves much fine work unfinished and a trail of profound desolation.

Additional Information and Declarations

Competing Interests

Author Contributions

Data Availability

The authors declare there are no competing interests.

Giase M. de Araujo performed the experiments, contributed reagents/materials/analysis tools, authored or reviewed drafts of the paper, approved the final draft.

Carlos A. Peres, Fabricio B. Baccaro and Rafael S. Guerta conceived and designed the experiments, analyzed the data, contributed reagents/materials/analysis tools, prepared figures and/or tables, authored or reviewed drafts of the paper, approved the final draft.

The following information was supplied regarding data availability:

The raw data are provided in Supplemental Information 1.

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
