# Peer review of "Urban waste disposal explains the distribution of Black Vultures (Coragyps atratus) in an Amazonian metropolis: management implications for birdstrikes and urban planning"

_PeerJ, doi:10.7717/peerj.5491_

## Round 0.1 · original submission · Major Revisions

It is clear that the three reviewers liked your manuscript. Most comments and suggestions were mostly made in order to improve even more your work. I ask you special attention to the Material and Methods section, where reviewers have raised important points that you need to take care of.

I hope to see a revised version of your manuscript very soon.

Good luck!

Reviewer 1 ·

Basic reporting

Please see 'general comments' below.

Experimental design

Please see 'general comments' below.

Validity of the findings

Please see 'general comments' below.

Additional comments

This is a worthy topic for research. The writing is reasonably good, although a few small corrections are noted. Likewise, most of the relevant references are included, but a few are missing and suggested. My main criticism is that more detail is needed in parts of the Methods, because I don’t think another researcher could reproduce your methods exactly with only the information presented here. Please see below for specific concerns.

• Abstract and elsewhere: Please change ‘aircrafts’ to ‘aircraft’.
• Line 57: The term ‘spatialized’ very odd; in fact I have not hear that word before. Please change to a more standard term.
• Lines 80-82: There are standard references you may want to cite regarding factors relating to hazard level: Dolbeer et al. (2000) Wildl Soc Bull 28:372-378; DeVault et al. (2011) Wildl Soc Bull 35:394-402.
• Line 97: The choice of references here is strange. Instead of Dolbeer & Wright (2008), you should cite the peer reviewed article Dolbeer (2006) J Wildl Manage 70:1345-1350. Also, the Dolbeer et al. (2011) strike report is quite old—these reports are produced annually and the most recent report should be cited instead.
• Line 135: Please clarify whether these 20 markets chosen were the only ones registered by the Manaus urban administration, or were they a sample of all those registered? If the latter, how did you choose the 20 markets to study?
• Methods—Bird Counts: This section needs much more detail. For example, how were the markets delineated? Did you record all vultures seen, or just those over or in the market? How did you avoid double counting? Were all areas where vultures could be visible from the car?
• Methods—Waste Storage: I would also like to see more detail here. It appears you recorded the proportion of waste type, but not the total amount of waste. So was a situation with 6 kg red meat out of 10 kg total waste categorized the same as a situation with 60 kg red meat out of 100 total kg waste? If so, that seems to be a poor way to score the amount of waste available.
• Methods—Historical Birdstrike Records and Sighting Reports: I understand how you gathered this information, but in the Methods you do not say why you gathered this information. What did you do with all the data you gathered?
• Lines 196-197: Please rephrase this sentence. I think you meant that all models with AIC < 2 were considered equally valid, but that is not what this sentence says.
• Results—I found myself looking for a table of the results (models), but this was not present. Please consider including this.
• Lines 271-272: Here you mention the ‘amount of fresh or decomposing animal waste available’, but I don’t think that is what you measured. See my comment above on proportion vs. amount.
• Line 333: Change ‘amble’ to ‘ample’.
• Conclusion—last two sentences are speculation and not warranted based on the findings from this paper.
• Figure 1: Where are the red squares? I couldn’t find any.

·

Basic reporting

Although English is not my first language, I believe the English used is quite adequate. The reading is clear, objective, succinct, fluid and technical.
References are adequate and mostly current. About 65% of references have less than 10 years, most of them less than 5 years. The oldest ones are either classic or indispensable references. Pay attention to the last listed reference (lines 521-522) - that is improperly positioned.
The manuscript is well structured. All divisions and subdivisions are adequate and the latter facilitate understanding. Figures and tables are appropriate and pertinent.
The results were well analyzed and support the conclusions presented.
The theme and objectives of the manuscript are consistent with the scope of the journal.

Experimental design

The purpose of the manuscript has scientific importance, both in terms of aircraft accidents involving birds such as public health.
Although it has been conducted in a large city in the Amazon region, it provides concrete, measurable and replicable data that can be extended to other urban centers, since the mentioned problems are recurrent in many urban centers around the world.
The methodology is well delineated and replicable.
.

Validity of the findings

The discussion is based on robust data, properly analyzed.

Additional comments

Congratulations. It was the best manuscript I have reviewed lately. I saw no problem whatsoever, which means that there was a great effort by you in preparing this manuscript.

·

Basic reporting

Basic Reporting
In general the manuscript is well written, but there are a few specific areas where this needs to be improved see below:
Line 57: Change ‘spatialized’ to ‘visual examined’.
Line 92: Remove ‘resting’.
Line 109: Change ‘in’ from before large Amazonian cities to ‘at’.
Line 112: Change ‘spatialized’ to ‘visual examined’.
Line 144: Change ‘modal’ to ‘main’.
Line 232: Remove ‘Delta’ and add the symbol for delta Δ.

Most of the citations are adequate, but here are a few suggestions:
Line 49: Vulture management is still an issue in cities that have stopped growing with adequate urban planning, so update the wording of this sentence.
Line 70: Dolbeer 2011 is specific to off-airport strikes, which often involve vultures. This should be updated to include information that bird strikes off airport property are increasing.
Line 72: Number of air carrier movements is not necessarily increasing in the states, so remove the Dolbeer 2005 reference.
Line 82: You should cite DeVault et al 2011 Interspecific Variation in Wildlife Hazards to Aircraft: Implications for Airport Wildlife Management to support this statement.
Line 93: The Novaes & Cintra 2013 reference deals with roosting sites not nesting, perhaps update this in the sentence.
Line 129: Add information on how much the city has grown over this time period.
Line 154: Can you add photographs of the storage containers?
Line 296: Are there any references about cultural notions of waste management in Brazil?
Figure 1 is not labeled properly. Please place the legend on the map itself (explaining red and blue squares). Why are there no red squares? Black arrows are not presence on the map, but white arrows are. Add an overview map of the cities location in Brazil. In the legend, add definitions to red circle symbols (sizes that correspond with how many vultures).

Experimental design

There are some fundamental flaws with the methodology.
First, how were the markets chosen? This is not stated. Were they randomly selected?
Next, I am concerned that vultures that were flying over the site within a 15 min survey period were counted. How far above the market were they? Were they low enough to be considered using the market, or were they just flying over. Since this is your response variable, I think it is important to make sure the reader is not confused. You could also subset observations to those vultures that were feeding or on the ground. For your surveys, how far was your radius of detection?
The composite index (DI) is not a useful addition to this manuscript. We do not know if markets located closer to airports are more hazardous than airports located further away. As vulture collisions occur at higher altitudes, perhaps markets that are further away from airports are considered more of a risk. I think this whole section should be removed.
The number of collisions reported in Table 2 are not specific to vultures, or other species that use markets, so the usefulness in the paper is limited.
Line 245: Can you add information about the height of the strikes? This would justify looking at vultures.
Line 258: What was the standard deviation of number of vultures observed?

Validity of the findings

There is some confusion on how data was collected, this should be fixed.
The conclusion section is a little misleading in that it does not discuss much about vulture abundance, and focuses on vultures as umbrella species. However, no other species were recorded in this study, so this statement is not appropriate.

---

## Round 0.2 · accepted · Accept

Congratulations on the Acceptance. Please accept my condolences over the loss of your co-author.

# Reviewer 1 ·

Basic reporting

no comment

Experimental design

no comment

Validity of the findings

no comment

Additional comments

Nice job with the revision.